# The effect of adjuvant therapy for patients with intrahepatic cholangiocarcinoma after surgical resection: A systematic review and meta-analysis

**Qiao Ke[1◐], Nanping Lin[1◐], Manjun Deng[1◐], Lei Wang[1,2]\*, Yongyi Zeng[1], Jingfeng Liu[1,3]**

**1** Department of Hepatopancreatobiliary Surgery, Mengchao Hepatobiliary Hospital of Fujian Medical University, Fuzhou, Fujian, PR, China, **2** Department of Radiation Oncology, Mengchao Hepatobiliary Hospital of Fujian Medical University, Fuzhou, Fujian, PR, China, **3** Liver Disease Center, The First Affiliated Hospital of Fujian Medical University, Fuzhou, Fujian, PR, China

◐ These authors contributed equally to this work.
\* wangleiy001@126.com

**Data Availability Statement:** All data generated or analyzed during this study are included in the published articles

## Abstract

### Background

Resection is still the only potentially curative treatment for patients with intrahepatic cholangiocarcinoma (ICC), but the prognosis remains far from satisfactory. However, the benefit of adjuvant therapy (AT) remains controversial, although it has been conducted prevalently. Hence, a meta-analysis was warranted to evaluate the effect of AT for patients with ICC after resection.

### Patients and methods

PubMed, MedLine, Embase, the Cochrane Library, Web of Science were used to identify potentially eligible studies from Jan.1st 1990 to Aug. 31st 2019, investigating the effect of AT for patients with ICC after resection. Primary endpoint was overall survival (OS), and secondary endpoints was recurrence-free survival (RFS). Hazard ratio (HR) with 95% confidence interval (CI) was used to determine the effect size.

### Results

22 studies with 10181 patients were enrolled in this meta-analysis, including 832 patients in the chemotherapy group, 309 patients in the transarterial chemoembolization (TACE) group, 1192 patients in the radiotherapy group, 235 patients in the chemoradiotherapy group, and 6424 patients in the non-AT group. The pooled HR for the OS rate and RFS rate in the AT group were 0.63 (95%CI 0.52~0.74), 0.74 (95%CI 0.58~0.90), compared with the non-AT group. Subgroup analysis showed that the pooled HR for the OS rate in the AT group compared with non-AT group were as follows: chemotherapy group was 0.57 (95%CI = 0.44~0.70), TACE group was 0.56 (95%CI = 0.31~0.82), radiotherapy group was 0.71 (95%CI = 0.39~1.03), chemoradiotherapy group was 0.73 (95%CI = 0.57~0.89),

**Funding:** This work was supported by Startup Fund for scientific research, Fujian Medical University (Grant number: 2018QH1195), but no authors in this study received a salary from this fund.

**Competing interests:** NO authors have competing interests

**Abbreviations:** ICC, intrahepatic cholangiocarcinoma; AT, adjuvant therapy; TACE, transarterial chemoembolization; LNM, lymph node metastasis; OS, overall survival; RFS, recurrence-free survival; HR, Hazard ratio; OR, odd ratio; CI, confidence interval.

positive resection margin group was 0.60 (95%CI = 0.51~0.69), and lymph node metastasis (LNM) group was 0.67 (95%CI = 0.57~0.76).

## Conclusion

With the current data, we concluded that AT such as chemotherapy, TACE and chemoradiotherapy could benefit patients with ICC after resection, especially those with positive resection margin and LNM, but the conclusion needed to be furtherly confirmed.

## Introduction

Intrahepatic cholangiocarcinoma (ICC) is the second most common primary liver cancer following hepatocellular carcinoma with a stably growing incidence and mortality[1, 2]. Surgical resection is still the most preferred treatment for patients with ICC, but only 15% of patients have the chance of surgery at initial diagnosis[3–5]. However, the prognosis of patients with ICC after resection remains far from satisfactory with the 5-year survival rate around 30%[6, 7]. Hence, concerns have always been focused on any strategies intended to improve the prognosis.

Various kinds of adjuvant therapies (AT), such as chemotherapy[8–10], radiotherapy[11, 12], transarterial chemoembolization (TACE)[13, 14], and chemoradiotherapy[15] have been conducted prevalently to improve the prognosis of patients after resection, and 21.4%-57.7% of patients were reported to receive AT after resection[14, 16]. However, the benefit of AT remains controversial[8, 9, 12]. Considering that randomized controlled trials or prospective studies evaluating the clinical vale of AT are hard to conduct, a comprehensive systematic review and meta-analysis is needed to confirm it.

## Material and method

This study was based on published studies and the informed consent of the patients and the ethical approval were not required. This meta-analysis was conducted according to the preferred Reporting Items for Systematic Reviews and Meta-Analyses (PRISMA).

### Literature search

A comprehensive search on the existing published medical literature was conducted by Qiao Ke and Nanping Lin to investigate the value of the AT for patients with ICC after surgical resection. English electronic databases such as PubMed, MedLine, Embase, the Cochrane Library, Web of Science were used to search the literature from Jan.1st 1990 to Aug. 31st 2019. Key words were as follows: ("intrahepatic cholangiocarcinoma" or "ICC" or "iCCA") AND ("adjuvant therapy" or "transarterial chemoembolization" or "chemotherapy or "radiotherapy" or "chemoradiotherapy"). Any potentially eligible studies were then identified manually through the references of the included studies, reviews, letters and comments.

### Selection criteria

**Inclusion criteria.** i) patients with ICC confirmed by pathology; ii) patients receiving surgical resection; iii) groups must include AT group and non-AT group; iv) outcomes must include the long-term outcomes.

**Exclusion criteria.** i) patients including gallbladder carcinoma or extrahepatic cholangio-carcinoma; ii) patients receiving neoadjuvant therapy; iii) patients receiving palliative resection; iv) data on the long-term outcomes was not available; v) studies based on overlapping cohorts deriving from the same center; vi) reviews, comments, letters, case report, and conference abstract.

Of note, considering that the data of most of the American studies came from the national cancer data base (NCDB), we only incorporated the study with longest research span if overlapping cohorts existed among studies.

## Intervention

Hepatectomy was conducted with or without lymph node dissection[17, 18], regardless of margin status.

AT was defined as any strategies administrated before recurrence, regardless of TACE, chemotherapy, radiotherapy, and chemoradiotherapy.

## Data extraction

Data such as the author's first name, year of publication, study methods, patient's characteristic, interventions, and outcomes were extracted and assessed by Qiao Ke and Nanping Lin with predefined forms. The hazard ratios (HRs) of OS or RFS were extracted directedly from the original data or extracted from the Kaplan-Meier curves according to the methods described in detail by Tierney et al[19]. and Parmar et al[20]. In case of disagreement, a third investigator, Manjun Deng, was intervened to reach a conclusion.

## Quality assessment

The quality of non-randomized studies was assessed by the modified Newcastle-Ottawa Scale (NOS)[21], and more than 7 stars were defined as high quality, 4~6 star as medium quality, and <4 stars as low quality.

## Statistical analysis

The meta-analysis was registered at http://www.crd.york.ac.uk/PROSPERO/ (Review registry 145810) and was performed using Stata 14. Considering all of the included studies were retrospective cohort studies, endpoints in this meta-analysis were evaluated by HRs and 95%CIs using the random-effects model[22, 23]. Subgroup analyses were conducted in the group of different AT strategies, R1 resection, and lymph node metastasis. Sensitivity analysis was conducted to observe whether the present result would be affected by any one study. Publication bias was determined using Begg's and Egger's tests, and "trim and fill" method was introduced to check the effect of potentially unpublished studies on the present result.

# Results

## Base characteristic of the included studies

Totally, 1267 records were excluded from the initially identified 1289 records. 22 studies including 23 cohorts and 10181 patients were enrolled in this meta-analysis[9, 11–14, 16, 24–39]. Groups were classified as follows: 832 patients in the chemotherapy group, 309 patients in the TACE group, 1192 patients in the radiotherapy group, 235 patients in the chemoradiotherapy group, and 6424 patients in the non-AT group. Of note, both adjuvant chemotherapy and adjuvant chemoradiotherapy were evaluated in Sur's study[33], so the former was defined as

Sur 2014a and the latter was defined as Sur 2014b. The search strategies and results were shown in Fig 1.

The characteristics and baseline demographic data of the patients in each research were listed in Table 1. Of note, two studies were international multi-centers ones[16, 33]. Details of AT in the included studies were depicted in Table 2. NOS score of each included study was exhibited in Table 3, among of which 20 studies were scored 7–9[9, 11–14, 16, 25–37, 39] and two were scored 5–6[24, 38].

## Endpoints

The OS and RFS comparing between AT group and non-AT group were evaluated in 22[9, 11–14, 16, 24–34, 36–39] and 6 included cohorts [14, 24, 27, 32, 35, 36], respectively. Using a random-effect model, the pooled HR for the OS and RFS in the AT group were 0.63 (95%CI 0.52~0.74, Fig 2A), and 0.74 (95%CI 0.58~0.90, Fig 2B), respectively, compared with the non-AT group.

## Subgroup analysis stratified by different AT strategies

The OS and RFS comparing between adjuvant chemotherapy group and non-AT group were evaluated in 9[9, 25, 29–31, 33, 36–38] and 2[35, 36] included cohorts, respectively. Using a random-effect model, the pooled HR for the OS and RFS in the AT group were 0.57 (95%CI 0.44~0.70, Fig 3A), and 0.75 (95%CI 0.45~1.05, Fig 3B), respectively, compared with the non-AT group.

The OS and RFS comparing between adjuvant TACE group and non-AT group were evaluated in 5[13, 14, 27, 28, 32] and 3[14, 27, 32] included cohorts, respectively. Using a random-effect model, the pooled HR for the OS and RFS in the adjuvant TACE group were 0.56 (95% CI 0.31~0.82, Fig 4A), and 0.74 (95%CI 0.55~0.93, Fig 4B), respectively, compared with the non-AT group.

The OS comparing between adjuvant radiotherapy group or adjuvant chemoradiotherapy group and non-AT group were evaluated in 4[11, 12, 26, 34] and 3[24, 33, 39] included cohorts, respectively. Using a random-effect model, the pooled HR for the OS in the adjuvant radiotherapy group and adjuvant chemoradiotherapy group were 0.71 (95%CI 0.39~1.03, Fig 5A), and 0.73 (95%CI 0.57~0.89, Fig 5B), respectively, compared with the non-AT group.

## Subgroup analysis stratified by high risk factors

The effect of AT on the patients with positive resection margin was evaluated in 4 included cohorts [16, 24, 25, 29]. Using a random-effect model, the pooled HR for the OS in the AT group was 0.60 (95%CI 0.51~0.69, Fig 6A), compared with the non-AT group. The effect of AT on the patients with LNM was evaluated in 4 included cohorts [9, 16, 24, 26]. Using a random-effect model, the pooled HR for the OS in the AT group was 0.67 (95%CI 0.57~0.76, Fig 6B), compared with the non-AT group.

## Sensitivity analysis

Sensitivity analysis was conducted in the primary endpoint comparing between AT group and non-AT group, and result showed that the pooled HR for the OS in the AT group did not change substantially after any study was removed compared with the non-AT group (Fig 7), which indicated that the present results in this study were robust.

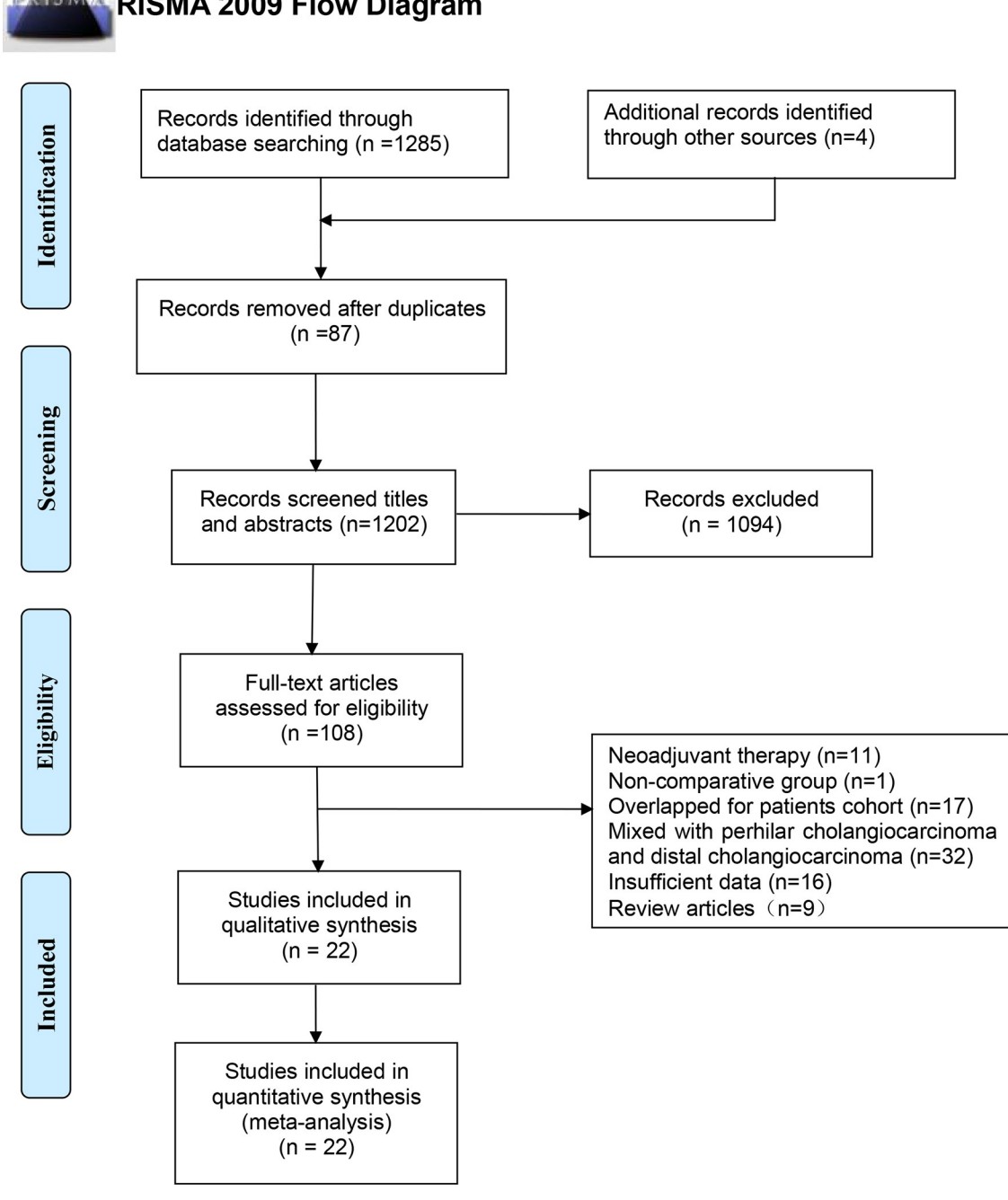

**RISMA 2009 Flow Diagram**

*From:* Moher D, Liberati A, Tetzlaff J, Altman DG, The PRISMA Group (2009). *P*referred *R*eporting *I*tems for *S*ystematic Reviews and *M*eta-*A*nalyses: The PRISMA Statement. PLoS Med 6(6): e1000097. doi:10.1371/journal.pmed1000097

For more information, visit **www.prisma-statement.org**.

**Fig 1. PRISMA flow diagram showing selection of articles for meta-analysis.**

**Table 1. Basic characteristics of included studies.**

| Study | Country | Design | Period | Primary endpoints | Sex(M/F) | LNM (+/-) | Vascular invasion (+/-) | Resection margin (+/-) | Follow-up (months) |
|---|---|---|---|---|---|---|---|---|---|
| Roayaie 1998[24] | US | Single center | 1991–1997 | OS/RFS | NR | 8/8 | NR | NR | 35.7(0.1–73.2) |
| Jan 2005[25] | China | Single center | 1997–2001 | OS | 128/184 | NR | NR | 222/90 | 14.1(1.05–167.6) |
| Jiang 2010[26] | China | Single center | 1998–2008 | OS | 52/38 | 90/0 | NR | NR | 13.2(0.3–123) |
| Shen 2011[27] | China | Single center | 2002–2003 | OS/RFS | 88/37 | 10/115 | 38/87 | NR | 18(3–96) |
| Wu 2012[28] | China | Single center | 2005–2006 | OS | 88/26 | 11/103 | 14/100 | NR | NR |
| Ribero 2012[30] | Italy | Multi-center | 1990–2008 | OS | 243/191 | 113/321 | 211/223 | NR | 36.5 |
| Bhudhisawasdi 2012 [29] | Thailand | Single center | 1998–2002 | OS | 116/55 | NR | 145/26 | 141/30 | NR |
| Li 2013[32] | China | Single center | 2000–2011 | OS/RFS | NR | 34/177 | 51/160 | NR | NR |
| Liu 2013[31] | China | Single center | 2005–2011 | OS | 48/33 | 50/31 | NR | NR | NR |
| Sur 2014[33] | US | Multi-center | 1998–2006 | OS | NR | 128/510 | NR | 180/458 | NR |
| Miura 2015[34] | US | Multi-center | 1998–2011 | OS | NR | NR | NR | NR | NR |
| Li 2015[14] | China | Single center | 2008–2011 | OS/RFS | 368/185 | 104/449 | 73/480 | NR | 25.3(2.2–76.2) |
| Okumaura 2016[36] | Japan | Single center | 2004–2015 | OS/RFS | 67/42 | 32/77 | 69/40 | 18/91 | NR |
| Luvira 2016[35] | Thailand | Single center | 2004–2009 | RFS | 26/24 | 18/32 | NR | 27/23 | NR |
| Hammad 2016[12] | US | Multi-center | 1998–2013 | OS | 819/755 | 607/967 | NR | NR | NR |
| Jeong 2017[13] | China | Single center | 2011–2015 | OS | 28/14 | 15/27 | 16/26 | NR | 36(11–65) |
| Tran 2017[39] | US | Multi-center | 2004–2012 | OS | NR | NR | NR | NR | NR |
| Schweitzer 2017[37] | Germany | Single center | 2000–2015 | OS | 111/86 | 45/152 | 44/153 | NR | NR |
| Reames 2017[9] | International | Multi-center | 1990–2015 | OS | 638/516 | 200/954 | 217/805 | 146/992 | NR |
| Zheng 2018[11] | China | Single center | 2007–2016 | OS | NR | 31/18 | 10/39 | NR | NR |
| Lee 2019[16] | US | Multi-center | 2004–2014 | OS | 1315/1498 | 582/2231 | NR | 649/2164 | 25.2(13.2–42) |
| Sahara 2019[38] | International | Multi-center | 1990–2015 | OS | NR | NR | NR | NR | 21.2(11.2–38.9) |

M: male; F: female; LNM: lymph node metastasis; NR: not report; OS, overall survival; RFS, recurrence-free survival.

## Publication bias analysis

Publication bias analysis was conducted in the primary endpoint comparing between AT group and non-AT group. Asymmetry was observed in the funnel plot (Fig 8) with significant publication bias in the egger's test (p = 0.004) but not in the Begg's test (p = 0.09). "Trim and

**Table 2. Interventions of adjuvant treatments in the included studies.**

| Study | Treatment types | Patients(yes/no) | Regimens |
|---|---|---|---|
| Roayaie 1998[24] | CRT | 9/7 | 5-FU(1000mg/m$^2$)+external beam radiotherapy (40-50GY) |
| Jan 2005[25] | CT | 118/194 | 5-FU+cisplatin+gemcitabine+doxorubicine+oxaliplatin |
| Jiang 2010[26] | RT | 24/66 | external beam radiation(50 Gy) |
| Shen 2011[27] | TACE | 53/72 | 5-FU (500 mg)/ carboplatin (100 mg)+iodized oil (3–5ml)+epirubicin (20 mg)+hydroxycamptothecin (10 mg) |
| Wu 2012[28] | TACE | 57/57 | 5-FU (500 mg)/ carboplatin (100 mg)+iodized oil (3–5ml)+epirubicin (20 mg)+hydroxycamptothecin (10 mg) |
| Ribero 2012[30] | CT | 116/318 | NR |
| Bhudhisawasdi 2012 [29] | CT | 54/117 | 5-FU(1000mg/m$^2$)+mitomycin C(10mg/m$^2$) |
| Li 2013[32] | TACE | 68/143 | 5-FU(500 mg)+iodized oil(3–5 ml)+epirubicin (20 mg)+hydroxycamptothecin (10 mg) |
| Liu 2013[31] | CT | 18/63 | 5-FU+cisplatin+gemcitabine+doxorubicine+oxaliplatin |
| Sur 2014a[33] | CT | 75/416 | NR |
| Sur 2014b[33] | CRT | 147/416 | NR |
| Miura 2015[34] | RT | 486/77 | NR |
| Li 2015[14] | TACE | 122/431 | 5-FU(500 mg)+iodized oil(3–5 ml)+epirubicin (20 mg)+hydroxycamptothecin (10 mg) |
| Okumaura 2016[36] | CT | 47/62 | Gemcitabine+ S-1 |
| Luvira 2016[35] | CT | 18/32 | 5-FU(1000mg/m$^2$)+mitomycin C(10mg/m$^2$) |
| Hammad 2016[12] | RT | 525/1049 | NR |
| Jeong 2017[13] | TACE | 9/33 | 5-FU+epirubicin+cisplatin |
| Tran 2017[39] | CRT | 79/170 | NR |
| Schweitzer 2017[37] | CT | 39/158 | gemcitabine (1000 mg/m$^2$)+cisplatin (25mg/m$^2$)+oxaliplatin (100mg/m$^2$) |
| Reames 2017[9] | CT | 347/807 | gemcitabine (1000 mg/m$^2$)+cisplatin (25mg/m$^2$)+oxaliplatin (100mg/m$^2$) |
| Zheng 2018[11] | RT | 26/23 | intensity-modulated radiotherapy(50-60Gy) |
| Lee 2019[16] | CT/CRT | 1189/1624 | NR |
| Sahara 2019[38] | RT | 131/505 | NR |

CRT: chemoradiotherapy; CT: chemotherapy; RT: radiotherapy; TACE, transarterial chemoembolization; NR: not report.

fill" analysis was then conducted, and 5 more studies were found to be potentially unpublished. The adjusted HR for the OS in the AT group was 0.73 (95%CI 0.63–0.85), compared with the non-AT group, indicating that the present result could not be affected by the unpublished studies.

## Discussion

The prognosis of patients with ICC after resection is still poor[5, 6], but the benefit of AT has always been questioned in clinical partly because the natural life span is too short and most of the patients have lost the chance of resection at diagnosis[3, 4]. Currently, with the advocation of LND and development of extended resection[40–42], the clinical value of AT should be re-evaluated. This is the first systematic review evaluating the clinical value of AT in the treatment of ICC comprehensively, which included 22 studies with 10181 patients, and results showed that patients could be benefited from AT in a whole. However, in our opinion, identifying the selected patients and choosing the appropriate AT strategy are the keys.

Chemotherapy is first to be administrated in the postoperative adjuvant treatments of ICC, and adjuvant chemotherapy is still the most preferred strategy in clinical up to now with the reported incidence as high as 46.6%[15]. However, the benefit of chemotherapy has been always been questioned mainly because cholangiocarcinoma is not sensitive to

**Table 3. Newcastle-Ottawa quality assessment of the included studies.**

| Study | Selection | | | | Comparability | Outcome | | | Scores |
|---|---|---|---|---|---|---|---|---|---|
| | Representativeness of the exposed cohort | Selection of the non-exposed cohort | Ascertainment of exposure | Outcome of interest was presented | | Assessment of outcome | Follow-up long enough for outcomes to occur | Adequacy of follow up of cohorts | |
| Roayaie 1998 [24] | ★ | | | ★ | ★ | ★ | ★ | | 5 |
| Jan 2005[25] | ★ | ★ | ★ | ★ | ★ | ★ | ★ | ★ | 8 |
| Jiang 2010[26] | ★ | ★ | ★ | | ★ | ★ | ★ | ★ | 7 |
| Shen 2011[27] | ★ | ★ | ★ | ★ | ★ | | ★ | ★ | 7 |
| Wu 2012[28] | ★ | ★ | ★ | | ★★ | ★ | ★ | ★ | 8 |
| Ribero 2012[30] | ★ | ★ | ★ | | ★ | ★ | ★ | ★ | 7 |
| Bhudhisawasdi 2012[29] | ★ | ★ | ★ | ★ | | ★ | ★ | ★ | 7 |
| Li 2013[32] | ★ | ★ | ★ | ★ | ★★ | ★ | ★ | ★ | 9 |
| Liu 2013[31] | ★ | ★ | ★ | ★ | ★ | ★ | ★ | ★ | 8 |
| Sur 2014[33] | ★ | ★ | ★ | | ★ | ★ | ★ | ★ | 7 |
| Miura 2015[34] | ★ | ★ | ★ | ★ | ★ | ★ | ★ | ★ | 8 |
| Li 2015[14] | ★ | ★ | ★ | ★ | ★★ | ★ | ★ | ★ | 9 |
| Okumaura 2016 [36] | ★ | ★ | ★ | ★ | ★ | ★ | ★ | ★ | 8 |
| Luvira 2016[35] | ★ | ★ | | ★ | ★ | ★ | | ★ | 6 |
| Hammad 2016 [12] | ★ | ★ | ★ | ★ | ★ | ★ | ★ | ★ | 8 |
| Jeong 2017[13] | ★ | ★ | ★ | ★ | | ★ | ★ | ★ | 7 |
| Tran 2017[39] | ★ | ★ | ★ | ★ | ★ | ★ | ★ | ★ | 8 |
| Schweitzer 2017 [37] | ★ | ★ | ★ | ★ | ★ | ★ | ★ | ★ | 8 |
| Reames 2017[9] | ★ | ★ | ★ | ★ | ★★ | ★ | ★ | ★ | 9 |
| Zheng 2018[11] | ★ | ★ | ★ | ★ | ★ | ★ | ★ | ★ | 8 |
| Lee 2019[16] | ★ | ★ | ★ | ★ | ★ | ★ | ★ | ★ | 8 |
| Sahara 2019[38] | ★ | ★ | ★ | ★ | | ★ | | ★ | 6 |

chemotherapeutics[43]. In this meta-analysis, adjuvant chemotherapy was confirmed to be associated with improved OS, which was coincident with the previous meta-analysis[10, 44]. In addition, Gemcitabine-based chemotherapy was confirmed to be superior to 5-Fu based chemotherapy in the improvement of prognosis[10, 44].

TACE is conducted widely in the management of ICC, such as adjuvant therapy for patients receiving resection[13, 14], and palliative treatment for unresectable ICC[45, 46]. However, someone argued the benefit of adjuvant TACE for ICC[27], mainly because ICC could metastasize specifically through lymph node. To the best of our knowledge, this is the first meta-analysis confirming the benefit of adjuvant TACE. Reasons are mainly due to that most of the recurrences are still intrahepatic ones[47], but it deserves further validation.

Radiotherapy is playing an increasing important role in the management of ICC with the development of stereotactic body radiotherapy[11]. From the other hand, metastatic lymph node is much more sensitive to radiotherapy[12]. However, the benefit of radiotherapy was not confirmed in this meta-analysis, which deserved our deep rethink. In addition, chemoradiotherapy is being more and more preferred in clinical, because synergistic effect is believed to between radiotherapy and chemotherapy[15]. This is the first meta-analysis identifying the

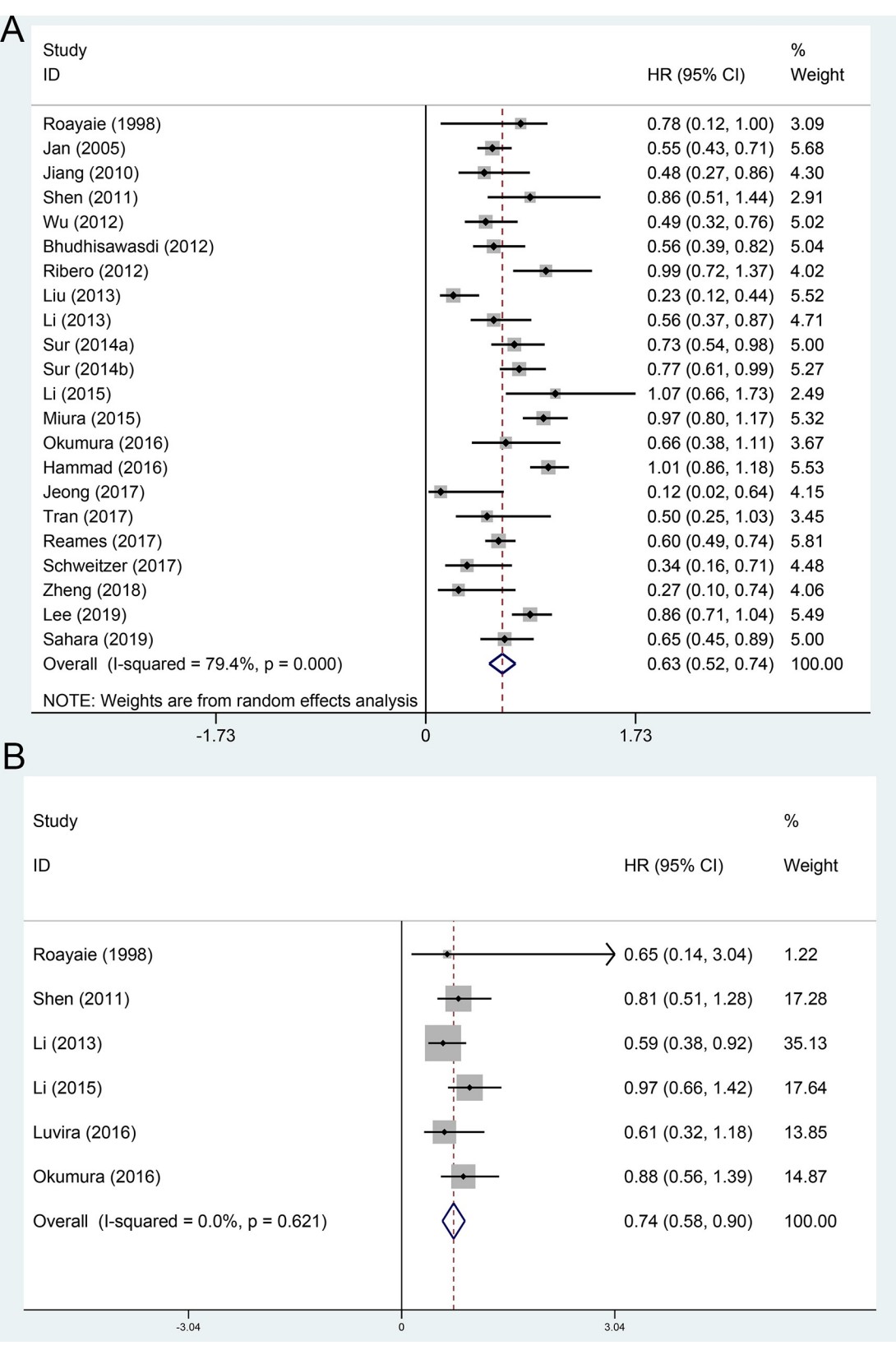

**Fig 2. Forest plot of the overall survival and recurrence-free survival rates between adjuvant therapy and operation only.** A, overall survival; B, recurrence-free survival.

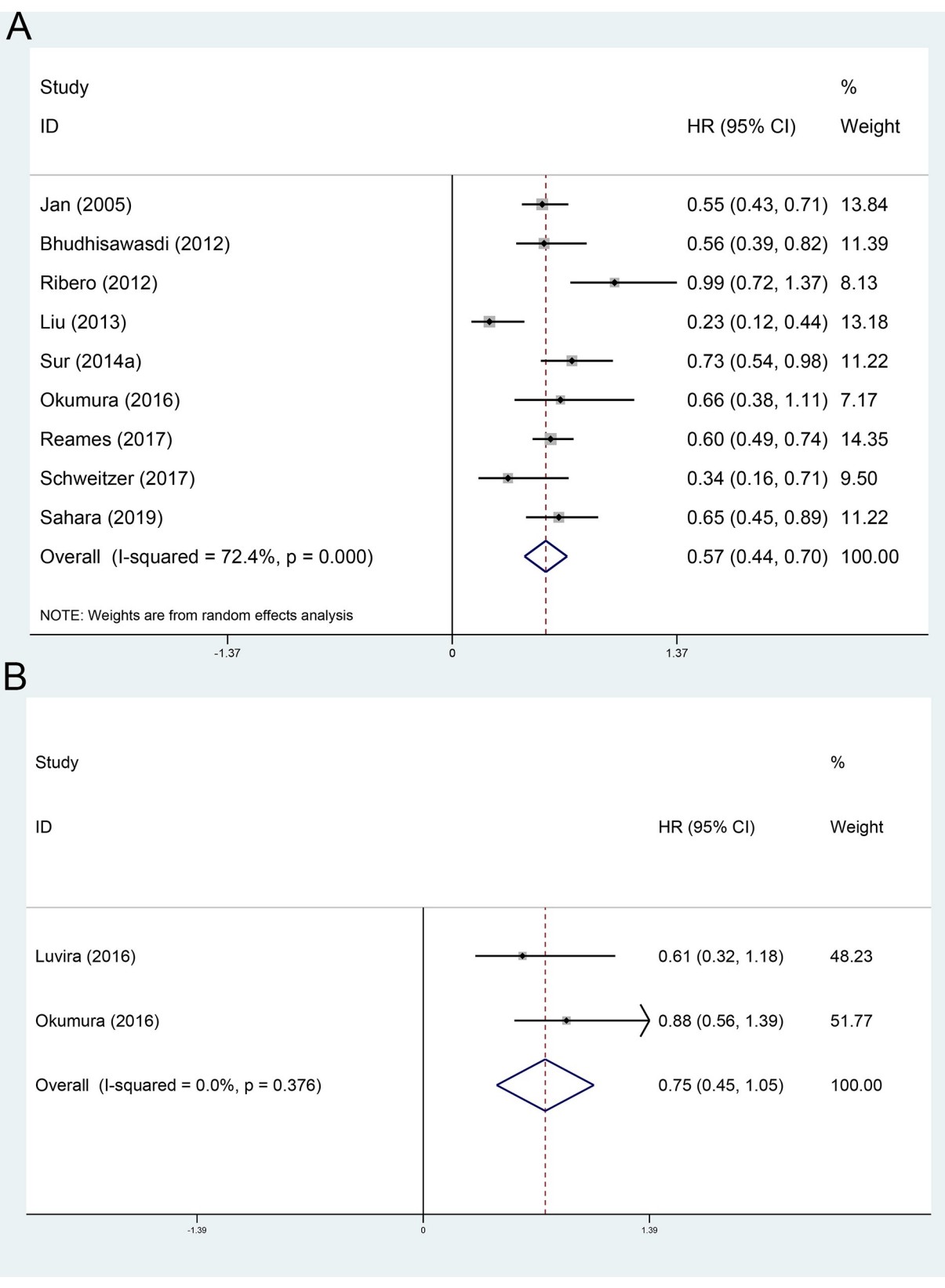

**Fig 3. Forest plot of the overall survival and recurrence-free survival rates between adjuvant chemotherapy and operation only.** A, overall survival; B, recurrence-free survival.

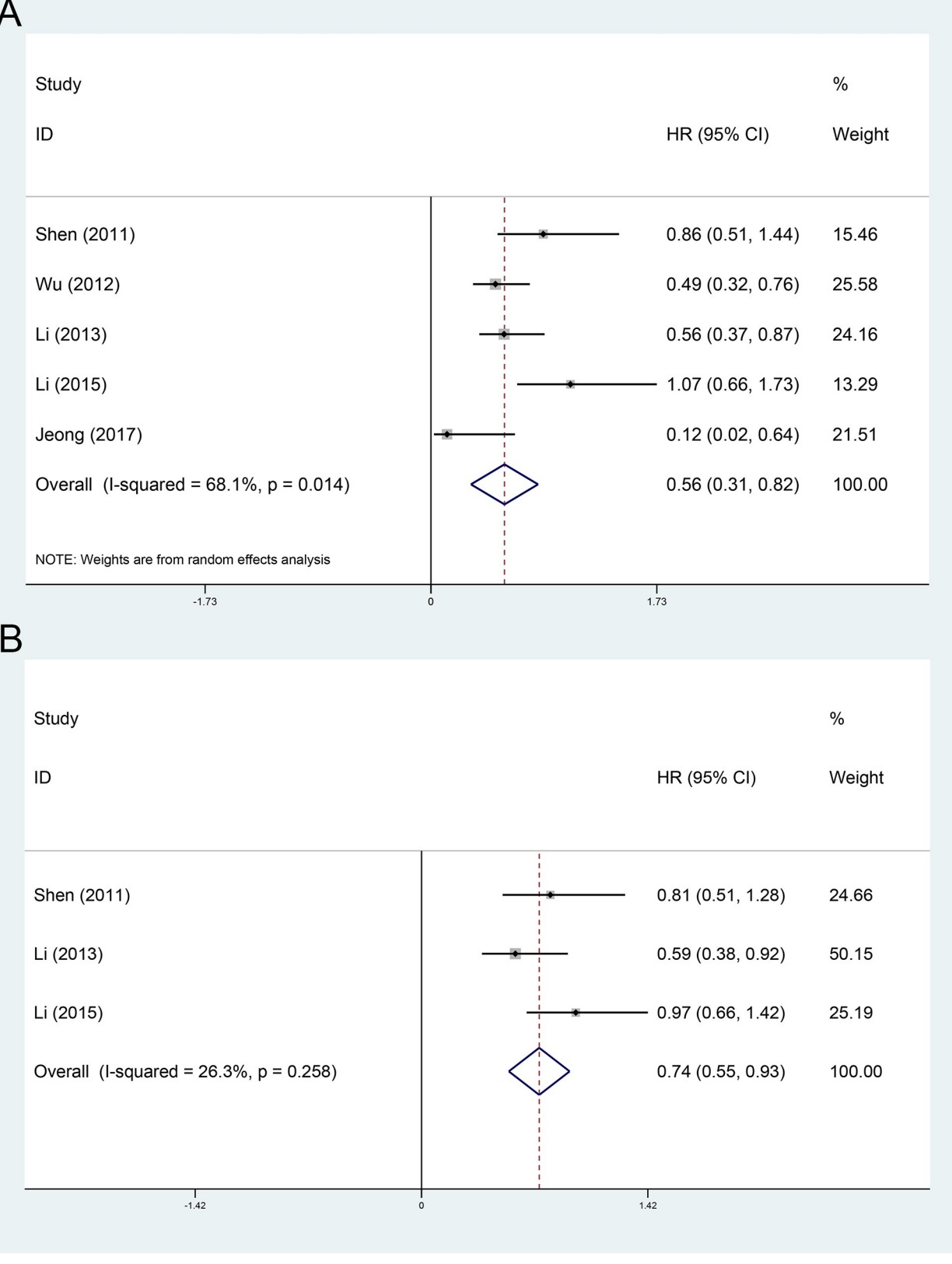

**Fig 4. Forest plot of the overall survival and recurrence-free survival rates between adjuvant TACE and operation only.** A, overall survival; B, recurrence-free survival.

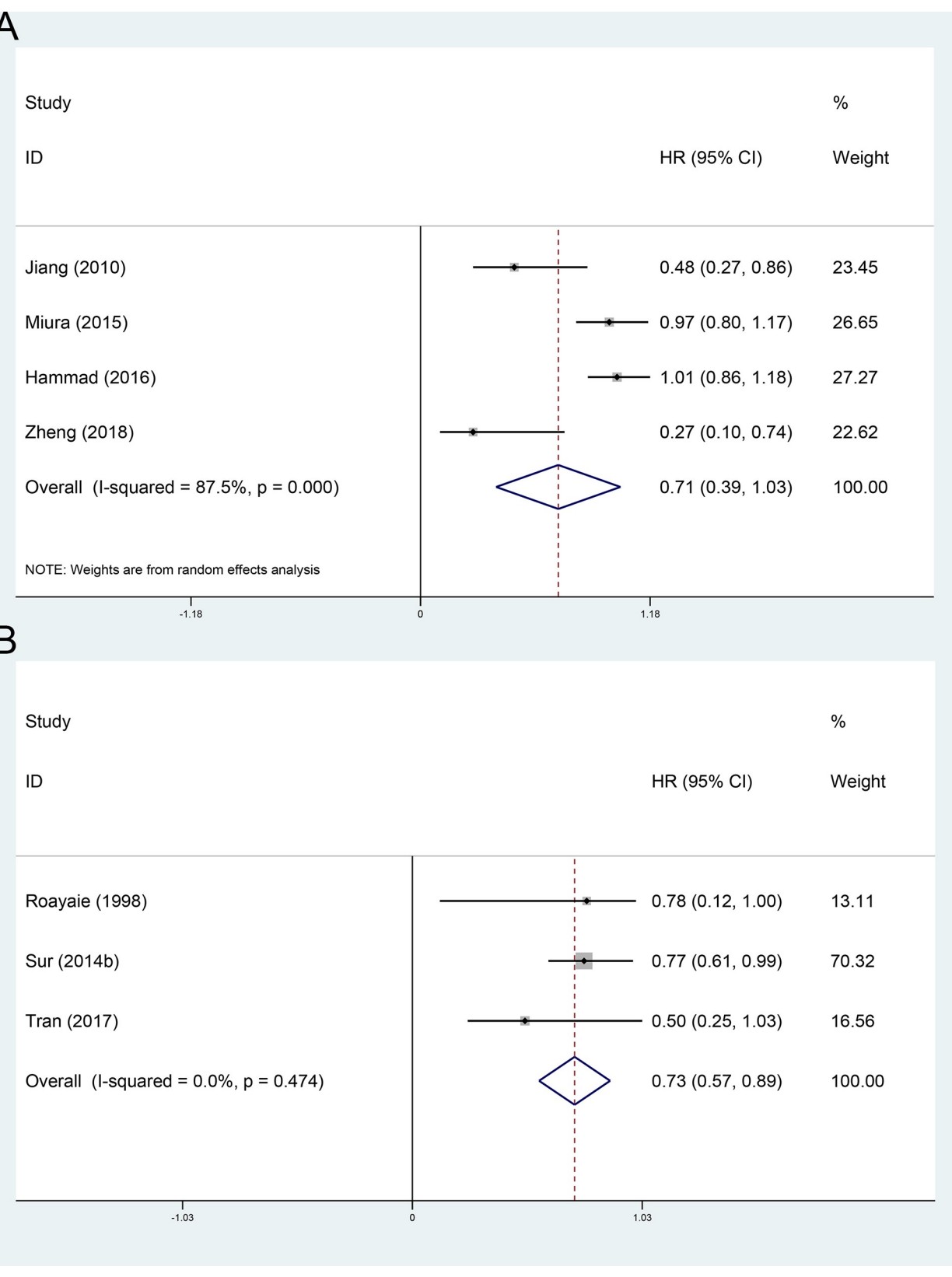

**Fig 5. Subgroup analysis of OS stratified by adjuvant radiotherapy and chemoradiotherapy.** A, adjuvant radiotherapy; B, adjuvant chemoradiotherapy.

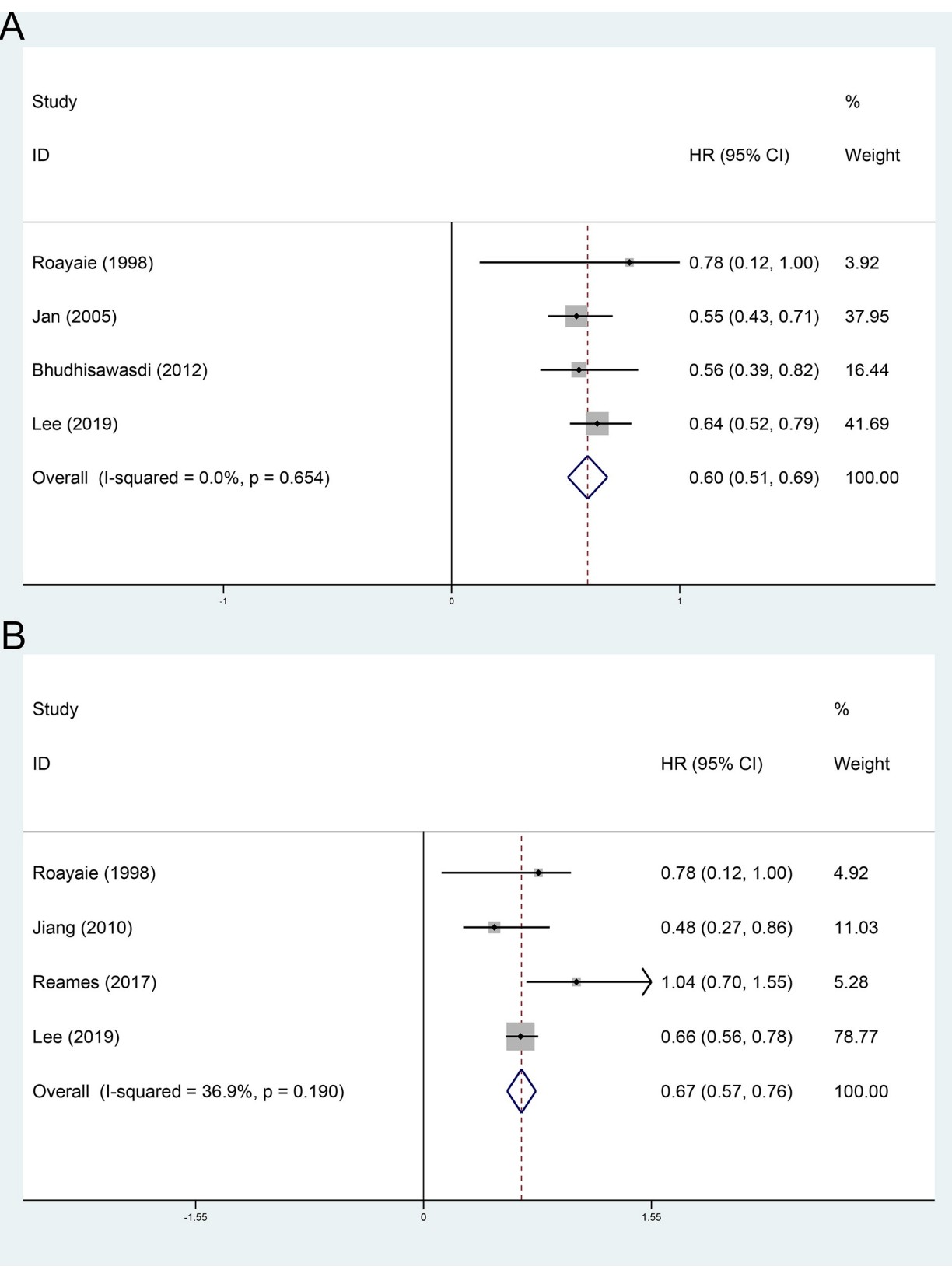

**Fig 6. Subgroup analysis stratified by high risk factors.** A, positive resection margin; B. lymph node metastasis.

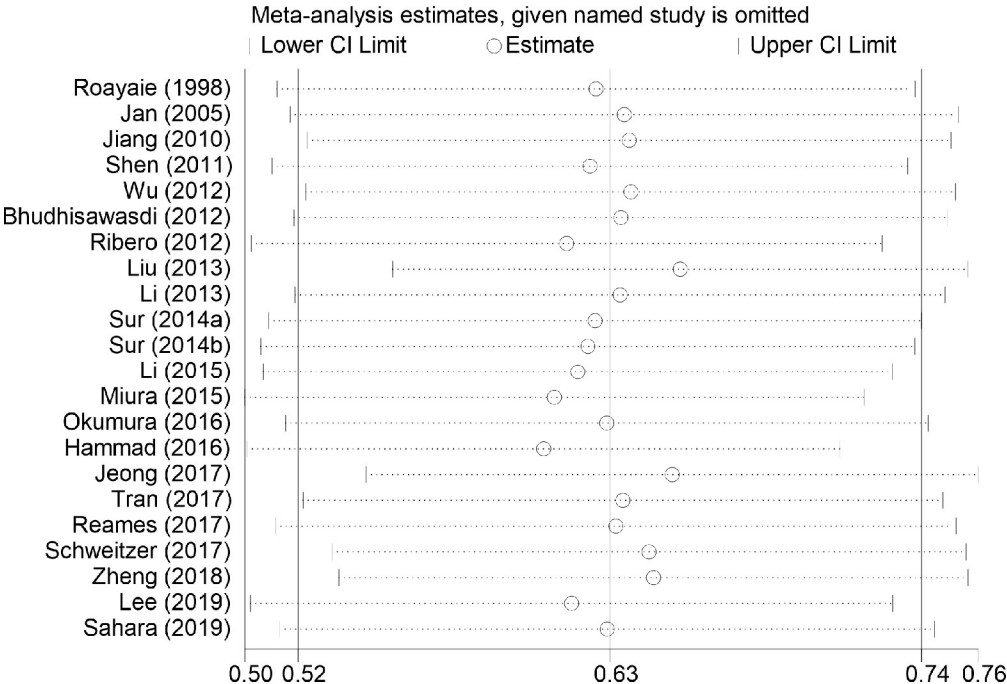

**Fig 7. Sensitivity analysis for overall survival in the included studies.**

benefit of adjuvant chemoradiotherapy, but either sequential or concurrent chemoradiotherapy deserves further study.

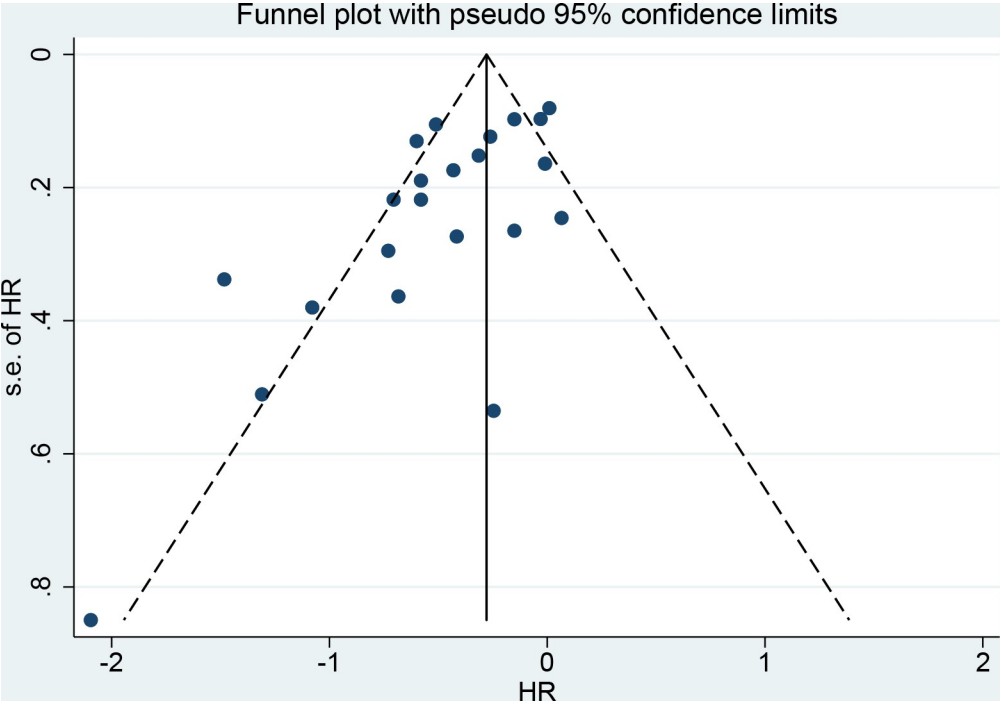

**Fig 8. Funnel plot of overall survival in the included studies.**

As is known to all, one size does not fit all, so identifying the selected patients who could be benefited from AT is also a big deal. Adjuvant chemotherapy and TACE are found to only benefit patients with "high risk", such as positive margins[24, 29], LNM[48], and advanced stages [16], but as for radiotherapy it is hard to say. Zheng et al[11] found that adjuvant radiotherapy could benefit patients with narrow surgical margin, but Hammad et al[12] reported that adjuvant radiotherapy could only improve the prognosis of patients with R0 resection rather than those with R1 resection and LNM. Hence, who would be benefited from AT, either with high risk or with low risk, is still a puzzle.

In the recent decades, pathway-targeted therapies made a rapid progress in solid tumors [49, 50]. Previous studies found that approximately 30%~40% of ICC patients exhibited actionable mutations, such as epidermal growth factor receptor (EGFR), and fibroblast growth factor receptor (FGFR) which shed light on the molecular targeted therapies on ICC[51, 52]. From the other hand, next-generation and exome sequencing studies found that 10%~15% of cholangiocarcinoma patients had DNA repair mutations[53], and 40% of cholangiocarcinoma patients had positive programmed cell death receptor 1 (PDL1) expression[54], who might be the potential beneficiaries of immunotherapies. Recent clinical trials have exhibited promising results in the advanced cholangiocarcinoma, which would change the trajectory of ICC management[55, 56]. In future, promises of adjuvant targeted therapies and/or immunotherapies have been expected in the on-going trials.

There were several restrictions of this meta-analysis. First, all the included studies were retrospective ones, indicating an obvious selection and recalling bias. Second, most of the studies were multi-centers or based on the database mainly due to the rare incidence of ICC, which meant that procedure of surgical resection and AT were different and bias was hard to avoid. Third, most of the cofounding factors such as radical resection and LNM were hard to be resorted in the original studies, which would weaken the conclusion. Fourth, RFS was evaluated in only six of the 22 included studies, which was insufficient to evaluate the effect of AT on recurrence. Fifth, considering that the span of the included studies was a little longer (1990~2019), during which the surgical techniques, chemotherapy agents and radiation approaches were different, our conclusion in this study deserved further validation. The last but not the least, publication bias was found in this meta-analysis, although the present result was found to be not changed after "trim and fill" analysis.

## Conclusion

With the current data, we concluded that AT would benefit patients with ICC after resection, but it deserved further validation. Considering that not all AT strategies would bring benefit to patients with ICC, and not all patients would be benefited from AT, identifying the potential beneficiaries of different AT is a priority in future.

## Supporting information

**S1 Table. PRISMA 2009 checklist.**
(DOC)

**S2 Table. The searching strategy of the pubmed database.**
(DOCX)

## Author Contributions

**Conceptualization:** Lei Wang, Yongyi Zeng, Jingfeng Liu.

**Data curation:** Qiao Ke, Nanping Lin, Manjun Deng.

**Formal analysis:** Qiao Ke, Nanping Lin, Manjun Deng, Lei Wang.

**Funding acquisition:** Lei Wang.

**Investigation:** Qiao Ke, Manjun Deng.

**Methodology:** Qiao Ke, Nanping Lin, Manjun Deng.

**Project administration:** Lei Wang, Yongyi Zeng, Jingfeng Liu.

**Resources:** Lei Wang, Jingfeng Liu.

**Software:** Qiao Ke.

**Supervision:** Yongyi Zeng, Jingfeng Liu.

**Validation:** Qiao Ke, Lei Wang.

**Visualization:** Lei Wang.

**Writing – original draft:** Lei Wang.

**Writing – review & editing:** Lei Wang.

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
