## [Decision Letter · Decision Letter 0]

30 Dec 2019

PONE-D-19-27317

The effect of adjuvant therapy for patients with intrahepatic cholangiocarcinoma after surgical resection: a systematic review and meta-analysis

PLOS ONE

Dear Dr. Lei Wang,

Thank you for submitting your manuscript to PLOS ONE. After careful consideration, we feel that it has merit but does not fully meet PLOS ONE’s publication criteria as it currently stands. Therefore, we invite you to submit a revised version of the manuscript that addresses the points raised during the review process.

We would appreciate receiving your revised manuscript within 60 days. To enhance the reproducibility of your results, we recommend that if applicable you deposit your laboratory protocols in protocols.io, where a protocol can be assigned its own identifier (DOI) such that it can be cited independently in the future. For instructions see: http://journals.plos.org/plosone/s/submission-guidelines#loc-laboratory-protocols

We look forward to receiving your revised manuscript.

Kind regards,

Gianfranco D. Alpini

Academic Editor

PLOS ONE

Journal Requirements:

2. At this time, we ask that you please provide the full search strategy and search terms for at least one database in the Supplementary information.

3. Thank you for stating the following beneath the Acknowledgments Section of your manuscript:

"This study was supported by Startup Fund for scientific research, Fujian Medical University (Grant number: 2018QH1195)."

"Not application"

Please provide an amended Funding Statement that declares *all* the funding or sources of support received during this specific study (whether external or internal to your organization) as detailed online in our guide for authors at http://journals.plos.org/plosone/s/submit-nowPlease state what role the funders took in the study.  If any authors received a salary from any of your funders, please state which authors and which funder. If the funders had no role, please state: "The funders had no role in study design, data collection and analysis, decision to publish, or preparation of the manuscript."

Reviewers' comments:

Reviewer's Responses to Questions

**Comments to the Author**

1. Is the manuscript technically sound, and do the data support the conclusions?

Reviewer #1: Yes

Reviewer #2: Yes

2. Has the statistical analysis been performed appropriately and rigorously? 

Reviewer #1: Yes

Reviewer #2: Yes

3. Have the authors made all data underlying the findings in their manuscript fully available?

Reviewer #1: Yes

Reviewer #2: Yes

4. Is the manuscript presented in an intelligible fashion and written in standard English?

Reviewer #1: Yes

Reviewer #2: Yes

5. Review Comments to the Author

Reviewer #1: I read the manuscript Ke et al with great interest. The authors tried to answer the question whether adjuvant therapy (in different forms) for intrahepatic CCA would be useful or not using the meta-analysis of the published paper.

With the large cohort of the meta-analysis and included study it looks like the statistical power seems adequate. However, the authors included studies which span over 30 years.

I have several comments;

1- Due to 30 years of span (1990-2019), as also can be seen by Table 2, some chemotherapy agents were different. Therefore, the response to chemotherapy could be different. This should be acknowledged in the discussion as the most limiting factor for chemotherapy and chemoradipotherapy. This long time span eventually impacts on the surgical techniques as well. The power of meta-analysis is the large cohort but different surgical techniques, (improved techniques most recently with improved diagnostics) change the overall survival of CCA patients. This should be also acknowledged as an important limitation.

2- page 3, line 70 - Varies should be Various

3- page 4 line 77, OMIT, "badly".

Again, the authors should add limitations to the study and soften the conclusion based on these limitations.

Reviewer #2: This manuscript by Qiao Ke et al. is a meta-analysis evaluating the effect of adjuvant therapy (AT) for patients with intrahepatic cholangiocarcinoma (ICC) after resection. Resection is still the only potentially curative treatment for patients with intrahepatic cholangiocarcinoma (ICC), but the prognosis remains far from satisfactory. The role of conventional therapies (transarterial chemoembolization, chemotherapy and radiotherapy) has yet to be fully defined, particularly in the adjuvant and second-line settings. This is the first systematic review evaluating the clinical value of AT in the treatment of ICC, including 22 studies with 10181 patients. Results showed that some patients could be benefited from AT in a whole. However, not all AT strategies would bring benefit to patients with ICC, and the benefit of adjuvant radiotherapy needs to be further validated. Identifying the selected patients and choosing the appropriate AT strategy is a major challenge for Clinics. The manuscript is well written and easy to follow. The topic of this manuscript is of great interest. The Reviewer has just some issues to address.

MAJOR

1. The major weaknesses of this study are well addressed by the Authors in the Discussion chapter: the studies are retrospective, indicating an obvious selection and recalling bias; most of the studies are multi-centers or based on the database mainly due to the rare incidence of ICC, which meant that procedure of surgical resection and AT were different and bias was hard to avoid; most of the cofounding factors such as radical resection and LNM are hard to be resorted in the original studies, which would weaken the conclusion; RFS is evaluated in only six of the 22 included studies, which is insufficient to evaluate the effect of AT on recurrence; publication bias is found in this meta-analysis.

2. Since a number of pathway-targeted therapies, as well as modulation of the immune environment, hold promise for patients with intrahepatic cholangiocarcinoma, the Authors should add this topic in the discussion.

6. PLOS authors have the option to publish the peer review history of their article (what does this mean?). If published, this will include your full peer review and any attached files.

Reviewer #1: No

Reviewer #2: No

---

## [Author Response · Author response to Decision Letter 0]

31 Jan 2020

Dear editor,

Firstly, we would like to thank the reviewers and the editor for the positive and constructive comments and suggestions. We have substantially revised our manuscript after reading the comments, and the revision was marked in red. In addition, the revision was re-edited literally, and the part of the method, especially the statistics, was checked repeatedly. 

Point by point responses to reviewers:

Journal Requirements:

Response: Thank you for your kindly reminder. We are sure that our manuscript is well meet the requirements of PLOS ONE.

2. At this time, we ask that you please provide the full search strategy and search terms for at least one database in the Supplementary information.

Response: Thank you for your professional comment. Full search strategy and search terms for PubMed was uploaded as the Supplementary file. However, manually searching was repeated conducted through the references of the included studies, reviews, letters and comments. 

3. Thank you for stating the following beneath the Acknowledgments Section of your manuscript:

"This study was supported by Startup Fund for scientific research, Fujian Medical University (Grant number: 2018QH1195)."

"Not application"

a.Please provide an amended Funding Statement that declares *all* the funding or sources of support received during this specific study (whether external or internal to your organization) as detailed online in our guide for authors at http://journals.plos.org/plosone/s/submit-now

b.Please state what role the funders took in the study.  If any authors received a salary from any of your funders, please state which authors and which funder. If the funders had no role, please state: "The funders had no role in study design, data collection and analysis, decision to publish, or preparation of the manuscript."

c.Please include your amended statements within your cover letter; we will change the online submission form on your behalf.

Response: Thank you for your kindly reminder. Lei Wang is the grantee of the funding, but "The funders had no role in study design, data collection and analysis, decision to publish, or preparation of the manuscript." We removed funding-related text from the manuscript and update our Funding Statement as you suggested.

Response: Thank you for your kindly reminder. We have added all the authors’s ORCID and updated our information.

Reviewer #1: I read the manuscript Ke et al with great interest. The authors tried to answer the question whether adjuvant therapy (in different forms) for intrahepatic CCA would be useful or not using the meta-analysis of the published paper.With the large cohort of the meta-analysis and included study it looks like the statistical power seems adequate. However, the authors included studies which span over 30 years.

I have several comments;

1- Due to 30 years of span (1990-2019), as also can be seen by Table 2, some chemotherapy agents were different. Therefore, the response to chemotherapy could be different. This should be acknowledged in the discussion as the most limiting factor for chemotherapy and chemoradipotherapy. This long time span eventually impacts on the surgical techniques as well. The power of meta-analysis is the large cohort but different surgical techniques, (improved techniques most recently with improved diagnostics) change the overall survival of CCA patients. This should be also acknowledged as an important limitation.

Response: Thank you for your professional comment. Firstly, we admitted that the span of the included researches was a little longer. Considering that most of the included studies were after 2010, we conducted a subgroup analysis stratified by the publication year before and after 2010 as you suggested, and results showed that the results of two subgroup analysis were coincident with the whole, which indicated that the conclusion was robust to some extent. Secondly, as you indicated that “some chemotherapy agents were different “, and the relevant subgroup analysis was published previously, which was depicted in the “Discussion”. And, the emphasis and innovation were the subgroup analysis stratified by different adjuvant strategies including chemotherapy, TACE, radiotherapy, and chemoradiotherapy. Thirdly, as you pointed out that “different surgical techniques, (improved techniques most recently with improved diagnostics) change the overall survival of CCA patients”, but relevant subgroup analysis such as laparoscopic vs. open hepatectomy, major vs. minor hepatectomy, hepatectomy with or without lymphadenectomy were unable to conduct due to most of the data was unavailable. However, these limitations were emphasized in the “Discussion” as you suggested. 

2- page 3, line 70 - Varies should be Various

Response: Thank you for kindly reminder, and we corrected it in the revised manuscript.

3- page 4 line 77, OMIT, "badly".

Response: Thank you for kindly reminder, and we corrected it in the revised manuscript.

Again, the authors should add limitations to the study and soften the conclusion based on these limitations.

Response: Thank you for kindly reminder, and we had softened our conclusion in the revised manuscript.

Reviewer #2: This manuscript by Qiao Ke et al. is a meta-analysis evaluating the effect of adjuvant therapy (AT) for patients with intrahepatic cholangiocarcinoma (ICC) after resection. Resection is still the only potentially curative treatment for patients with intrahepatic cholangiocarcinoma (ICC), but the prognosis remains far from satisfactory. The role of conventional therapies (transarterial chemoembolization, chemotherapy and radiotherapy) has yet to be fully defined, particularly in the adjuvant and second-line settings. This is the first systematic review evaluating the clinical value of AT in the treatment of ICC, including 22 studies with 10181 patients. Results showed that some patients could be benefited from AT in a whole. However, not all AT strategies would bring benefit to patients with ICC, and the benefit of adjuvant radiotherapy needs to be further validated. Identifying the selected patients and choosing the appropriate AT strategy is a major challenge for Clinics. The manuscript is well written and easy to follow. The topic of this manuscript is of great interest. The Reviewer has just some issues to address.

MAJOR

1. The major weaknesses of this study are well addressed by the Authors in the Discussion chapter: the studies are retrospective, indicating an obvious selection and recalling bias; most of the studies are multi-centers or based on the database mainly due to the rare incidence of ICC, which meant that procedure of surgical resection and AT were different and bias was hard to avoid; most of the cofounding factors such as radical resection and LNM are hard to be resorted in the original studies, which would weaken the conclusion; RFS is evaluated in only six of the 22 included studies, which is insufficient to evaluate the effect of AT on recurrence; publication bias is found in this meta-analysis.

Response: Thank you for your professional comments, and these limitations above were hardly avoided, and would weaken the conclusion of this study. Hence, we had softened the conclusion in the revised manuscript.

2. Since a number of pathway-targeted therapies, as well as modulation of the immune environment, hold promise for patients with intrahepatic cholangiocarcinoma, the Authors should add this topic in the discussion.

Response: Thank you for your professional comment. Considering the great advances in the pathway-targeted therapies and immunotherapies, prognosis of patients with intrahepatic cholangiocarcinoma following resection would be improved. Hence, we checked the latest reports on this topic, and added the progress in the revised “Discussion”.

We are looking forward to receive your letter, and please contact me without any hesitation if you have any question.

Best Regards,

Yours sincerely, 

Qiao Ke

Corresponding to Lei Wang, E-mail: wangleiy001@126.com

---

## [Decision Letter · Decision Letter 1]

4 Feb 2020

The effect of adjuvant therapy for patients with intrahepatic cholangiocarcinoma after surgical resection: a systematic review and meta-analysis

PONE-D-19-27317R1

Dear Dr. Lei Wang,

We are pleased to inform you that your manuscript has been judged scientifically suitable for publication and will be formally accepted for publication once it complies with all outstanding technical requirements.

With kind regards,

Gianfranco D. Alpini

Academic Editor

PLOS ONE

Additional Editor Comments (optional):

Reviewers' comments:

Reviewer's Responses to Questions

**Comments to the Author**

1. If the authors have adequately addressed your comments raised in a previous round of review and you feel that this manuscript is now acceptable for publication, you may indicate that here to bypass the “Comments to the Author” section, enter your conflict of interest statement in the “Confidential to Editor” section, and submit your "Accept" recommendation.

Reviewer #1: All comments have been addressed

Reviewer #2: All comments have been addressed

2. Is the manuscript technically sound, and do the data support the conclusions?

Reviewer #1: Yes

Reviewer #2: Yes

3. Has the statistical analysis been performed appropriately and rigorously? 

Reviewer #1: Yes

Reviewer #2: Yes

4. Have the authors made all data underlying the findings in their manuscript fully available?

Reviewer #1: Yes

Reviewer #2: Yes

5. Is the manuscript presented in an intelligible fashion and written in standard English?

Reviewer #1: Yes

Reviewer #2: Yes

6. Review Comments to the Author

Reviewer #1: Authors responded to all my comments properly and appropriate changes were made. No further comments.

Reviewer #2: The Authors answered satisfactorily to reviewers' comments.

7. PLOS authors have the option to publish the peer review history of their article (what does this mean?). If published, this will include your full peer review and any attached files.

Reviewer #1: No

Reviewer #2: No

---

## [Editor Report · Acceptance letter]

7 Feb 2020

PONE-D-19-27317R1 

The effect of adjuvant therapy for patients with intrahepatic cholangiocarcinoma after surgical resection: a systematic review and meta-analysis 

Dear Dr. Wang:

I am pleased to inform you that your manuscript has been deemed suitable for publication in PLOS ONE. Congratulations! Your manuscript is now with our production department. 

With kind regards,

on behalf of

Dr. Gianfranco D. Alpini 

Academic Editor

PLOS ONE